# Protective Factors in the LGBTIQ+ Adolescent Experience: A Systematic Review

**DOI:** 10.3390/healthcare12181865

**Published:** 2024-09-16

**Authors:** Ruth A. Ancín-Nicolás, Yolanda Pastor, Miguel Ángel López-Sáez, Lucas Platero

**Affiliations:** Department of Psychology, Rey Juan Carlos University, Avda/Atenas, s/n, 28922 Alcorcón, Madrid, Spain; yolanda.pastor@urjc.es (Y.P.); miguel.lopez.saez@urjc.es (M.Á.L.-S.); lucas.platero@urjc.es (L.P.)

**Keywords:** protective factors, adolescence, LGBTIQ+, mental health, well-being

## Abstract

*Background*: LGBTIQ+ adolescents face multiple forms of violence due to minority stress, putting their mental health at risk. These adolescents, particularly trans and gender-diverse youth, face significant challenges, including family violence, bullying, and limited access to health services. This article investigates the factors that protect their mental health. *Method*: A systematic review of articles published between 2019 and 2024 was conducted using the PRISMA method. *Results*: The study identifies family and school support, peer relationships, and individual resilience as key protective factors to reduce depression, anxiety, and suicidal behavior, as well as increase self-esteem and well-being. *Conclusions*: This review calls for further research into protective factors related to the microsystem, exosystem, and macrosystem and the impact of intervention programs with adolescents and their families.

## 1. Introduction

The past four decades have witnessed substantial growth in support for LGBTIQ+ (lesbian, gay, bisexual, trans, intersex, and queer) rights, as reflected by the attention paid to this topic in the academic literature [1] (p. 371). LGBTIQ+ adolescents face significant violence, with 60–70% experiencing bullying, homelessness, and health service access issues, with a particular impact on intersex, trans, non-binary, and gender-diverse individuals, one in three of whom have contemplated suicide [2]. Being LGBTIQ+ increases stress in a society that favors heterosexuality, cisnormativity, and other hegemonic identities [3], especially for trans and gender diverse individuals [4,5].

Identity development in LGBTIQ+ adolescents differs from that of their cis, heterosexual, and endo peers, as they must identify their own gender and sexuality despite negative stigma in contexts in which often they lack information and affirmative resources. These youngsters also have to handle the decision about disclosing their non-normative identity and find peers [6]. These adolescents face a higher risk of auto-stigmatization and fewer protective factors, such as social support or health care access, than their heterosexual, cisgender peers [7]. The minority stress model [8] reveals how the distress experienced by these adolescents results from a majority social system that does not support dissenters. Distal stressors are part of the external violence they face, while proximal stressors refer to internalized rejection. These proximal stressors are especially relevant in identity development, particularly during a developmental stage of adolescence, where socialization plays a key role in the construction of self-concept and self-esteem [9]. The academic literature has identified several connections between the components of the self and a negative self-assessment among LGBTIQ+ youth, compared to their heterosexual peers [6,9,10].

LGBTIQ+ adolescents also report higher levels of stress, anxiety, depression, and alcohol and drug abuse, as well as more suicidal ideation and attempted suicide than their cisheterosexual peers [3,11]. Although these data are striking, risk factors should not be conflated with the impact of violence on LGBTIQ+ individuals themselves [5,12]. These adolescents also develop specific skills and coping mechanisms [13], problem-solving abilities that can be a major predictor of mental health in adulthood [14]. Therefore, being LGBTIQ+ does not automatically translate into worse health outcomes [5]. To better understand this phenomenon, this article investigates the health factors that shield LGBTIQ+ children from experiencing negative outcomes that result from their sexuality and gender. 

The risk factors identified with LGBTIQ+ youth have been extensively studied in the academic literature. When present, these factors escalate the likelihood of disorder manifestations, such as depression, anxiety, non-suicidal self-injury (NSSI), suicide, substance abuse, or poor sexual and reproductive health [15]. Health factors, on the other hand, are characteristics related to the individual or the environment that highlight strengths and shield individuals from harm related to stigmatizing environments, minimize suffering [16], and lessen the chance that risk factors will materialize [17]. Thus, health factors are supportive resources that serve as buffers against the consequences of violence that targets adolescents, such as suicide, depression, loneliness, substance abuse, and sexually transmitted diseases [11]. The phenomena known to be protective factors are those experiences that, in the face of potentially harmful events, improve favorable childhood outcomes [18]. Youth-adult connectivity, school connectedness, and community involvement have all been linked to overall enhanced health and well-being in young people [19]. Health factors also include personal resilience and self-care, self-esteem, pride and self-acceptance, and access to supportive friends, family, and community, such as gay-straight alliances (GSA), affirming care, acceptance, and positive connections with LGBTIQ+ communities [3,15,20]. These factors reduce victimization and are predictors of overall health and well-being for LGBTIQ+ individuals [21]. Despite these findings, there is a notable gap in the literature that emphasizes protective factors—elements that not only prevent adverse outcomes, like suicide or depression, but also enhance overall quality of life and boost self-esteem. The goal of this study is to shed light on these protective factors, highlighting their significance and the need for more specific attention, both in academia and in public policies. Moreover, the research that has begun to emerge on protective factors is often interested in outcomes in adults, but not in adolescents. 

Adolescence is a concept that can be historicized [22] (p. 30), and although it is chronologically delimited, there is no consensus about the ages included in this stage. The World Health Organization [23] and United Nations [24] define adolescence as the period between 10 and 19 years of age, while other authors define adolescence as between 12 and 17 [22] (p. 30). For this article, we define adolescence as between 12 and 18 years old, since this age period covers the biological and social changes that have traditionally defined the teenage years. 

This article conducts a systematic review of the empirical literature on health factors among LGBTIQ+ people, organized based on a socioecological model, with the aim of contributing to future research on health and protective factors among LGBTIQ+ adolescents. The research question is: What are the protective factors that improve the mental health and well-being of LGBTIQ+ adolescents? The hypotheses, in turn, are: (1) the main protective factor for LGBTIQ+ adolescent mental health is social support; (2) the most important protective factors are family, school, and peers; and (3) the individual protective factors are self-esteem and self-efficacy. 

## 2. Materials and Methods

A systematic review approach was employed in this analysis, which comprised finding, assessing, and interpreting pertinent studies on the protective factors related to mental health in LGBTIQ+ adolescents, published between 2019 and 2024. The aim of a systematic review is to assemble all the empirical evidence that matches predetermined qualifying criteria, employing methodical techniques, and produce accurate outcomes [25]. The procedure described by Barbara Kitchenham and Stuart Charters [26] and the Preferred Reporting Items for Systematic Reviews and Meta-Analyses (PRISMA) standards were adopted for this review [27]. This review has not been registered.

### 2.1. Eligibility Criteria

The papers had to meet a number of criteria to be included in the study: (1) being published between 2019 and 2024 in a peer-reviewed journal in English and/or Spanish; (2) analyzing samples or subsamples of adolescents who identify as LGBTIQ+, aged 12 to 18; (3) identifying at least one protective factor associated with well-being or mental health; and (4) being either qualitative or quantitative. Qualitative studies had to report on at least one protective factor for LGBTIQ+ adolescents, while quantitative studies needed to consider the relationship between theoretical protective factors and mental health outcomes, highlighting statistically significant results (*p* < 0.05).

Since our focus is the mental health of LGBTIQ+ adolescents, articles on protective factors for problem behaviors such as drug use or sexual risk were excluded. Additionally, articles that did not include adolescents in their studies or did not discuss adolescents apart from adults, as well as those that only examined risk factors in relation to LGBTIQ+ adolescent health or behavior, were also excluded. Other systematic reviews were not included, nor were single case studies. Finally, gray literature was excluded, as only published and peer-reviewed studies were considered.

### 2.2. Information Sources and Search Study

The search was intentionally broad to minimize the risk of overlooking potentially relevant studies. The search strategy was developed for the concepts of adolescence, LGBTIQ+, and protective factors. The search strategies were created by three authors using a combination of subject headings and keywords, which are presented in Figure 1. Keyword selection was reviewed and developed further by a fourth author to ensure its relevance to the research question.

Typically, the research to date has focused on understanding the risk factors for the LGBTIQ+ population based on minority stress [8,28]. These studies are important for changing perspectives and analyzing the factors that improve the mental health and well-being of LGBTIQ+ youth, emphasizing what to promote rather than what to avoid when intervening with this population. Protective factors play a crucial role, not only in averting negative mental health outcomes (lowering depression and anxiety, etc.), but also in fostering resilience and coping mechanisms, well-being, and positive self-perception. These factors contribute to a holistic approach to mental health, which is equally important as the strategies aimed at reducing risks. By bringing these protective factors to the forefront, we hope to encourage further research and discussions that can lead to more comprehensive mental health interventions.

### 2.3. Study Selection Process

Once the inclusion criteria were determined, the titles, abstracts, and keywords were scanned against the eligibility criteria to identify potentially relevant articles. Relevant articles were obtained and entered into Microsoft Excel. Using Excel, duplicates were identified and removed. Two authors (R.A.A-N and L.P.) then performed title and abstract screening concurrently and independently, with conflicts resolved by a third author (Y.P-R). Articles that were unclear with regard to the inclusion and exclusion criteria were analyzed and selected upon the consensus of all the authors. The same screening process was used for the full-text articles.

### 2.4. Study Quality Assessment

An analysis of the quality of the papers was conducted by two authors (R.A.A-N and L.P.) working independently. The critical appraisal tool [29] was used to select papers with a qualitative methodology, and quality assessment criteria were used for articles with a quantitative methodology [30]. Both checklists were employed for the mixed methodology (Appendix A). Disagreements in the risk of bias scoring were resolved by a discussion involving all the authors.

### 2.5. Data Extraction and Analysis Procedure

A meta-analysis was not conducted due to the heterogeneity of the populations, the variability in the measurement tools used to assess protective factors, and the limited sample sizes observed in certain studies. A scoping review analysis was used to summarize the results [31]. To describe the studies in which the researchers studied protective factors for mental health in LGBTIQ+ adolescents, we used a thematic framework that included: (1) study data such as design, location, and sampling strategy; (2) sociodemographic characteristics of the sample; (3) protective factors; and (4) consequences or benefits of protective factors on mental health. The protective factors were organized according to Bronfenbrenner’s socioecological model, as proposed by Johns et al. [32] (see Figure 2). Microsoft Excel (version 2016) was used to create a standardized data extraction form from the outlined studies. Lastly, to characterize the investigations and findings, data from each cell were examined.

## 3. Results

### 3.1. Study Selection

The study process and selection can be seen in Figure 3. 

### 3.2. Characteristics of the Included Studies

The studies included were published between 2019 and February 2024, with 80% conducted in the United States and 4.3% in Australia and the United Kingdom. The remaining locations (Brazil, France, China, Europe, not reported) each accounted for 2.17% of the selected studies. Articles with qualitative methodology used semi-structured interviews, in-depth interviews, or focus groups. Articles with quantitative methodology used self-report questionnaires, either on paper or online. Finally, mixed-method articles used questionnaires and open-ended questions. The articles sampled incorporated an age range from 8 to 32, and although not all the life stages were represented, all the studies addressed adolescence.

### 3.3. Quality Assessment of the Studies

The scale for qualitative studies, the critical appraisal tool (Appendix A), had 22 items with 3 response types (“yes”, “no”, “don’t know”). The qualitative studies included had at least 19 “yes” responses [29]. Regarding the quantitative scale, the quality assessment criteria had 14 items with 4 response types (“yes”, “partial”, “no”, and “not applicable”). As 3 items were not applicable for all the studies selected, the final scale contained 11 items. The weighted scores of the included items had at least ≥0.70 out of 1 [30].

### 3.4. Protective Factors

Quantitative studies were the most common, with 35 articles, while qualitative methods were used in 6 articles and mixed methods in 5. Table 1, Table 2, Table 3 and Table 4 present the results of the articles numbered 1–46. Following the social-ecological model, the main factors studied were related to social support. The most studied social support was family support, including good communication and family acceptance, and family support was found to reduce the risk of mental health issues and increase self-esteem and well-being. School support, peer relationships, and gay-straight alliances also played crucial roles in reducing anxiety, depression, and bullying and increasing well-being and self-esteem. Microsystem factors such as spirituality, self-care, self-acceptance, pride, resilience, and conflict resolution skills were associated with increased self-esteem and general well-being (see Table 4).

Although they received less attention in the studies, exosystem and macrosystem factors highlighted not only the importance of access to positive therapy and medical treatment, but also the existence of supportive policies and training for teachers. These factors reduced depression and increased well-being and a sense of belonging. The studies also indicated that LGBTIQ+ adolescents have fewer protective factors when compared to their cisheterosexual counterparts. 

## 4. Discussion

This study reviews the protective factors that improve the mental health and well-being of LGBTIQ+ adolescents, based on the social-ecological systems framework. The results of the 46 items studied will be discussed based on the social-ecological systems categories. These results indicate several critical questions for future studies, social intervention, and policymaking. Overall, the findings underscore the fact that protective factors for LGBTIQ+ mental health and well-being are not static, but rather the result of a wider range of dynamically interacting elements arising from the individual, societal, organizational, policy, and socio-ecological levels of society. 

Regarding the mesosystem, most of the articles studied social support as a protective factor, and the first hypothesis—“the main protective factor for LGBTIQ+ adolescent mental health is social support”—is upheld. Within social support, most articles emphasized family support, particularly from parents, as a protective factor, showing that individuals who feel connected and supported by their families exhibit greater well-being and acceptance [20,50], less anxiety and depression [50,57], and lower suicidal ideation [12], especially for bisexual and non-binary adolescents [34]. The next most studied factor was school, where peer support and GSAs have been found to increase well-being [47] and connectedness [51], as well as reduce bullying and depression [34,36], anxiety [57], NSSI, and suicide [45]. The last factor in the mesosystem was peers, where peer support increases well-being [20], reduces suicide and NSSI [53], and decreases bullying [3]. In contrast to their cissexual counterparts, LGBTIQ+ individuals experience systemic and interpersonal discrimination [7]. Understanding the need for social support for this population’s well-being, stress management, and personal growth is crucial. Anti-LGBTIQ+ contexts consistently produce discomfort, lower life satisfaction, and detrimental effects on individual development. In contrast, affirming contexts increase the positive effects [28,74]. Additionally, building connections with LGBTIQ+ allies, who listen to them and offer emotional support, emerges as a crucial undertaking for healthy psychological self-development [74]. These findings are supported by the recent literature, which indicates that adolescents who feel listened to in their environments (by family members, school staff, and peers) feel empowered and recognized [75,76]. These results support the second hypothesis: “the most important protective factors are family, school, and peers”.

Microsystem factors were the next most studied determinants. Within this category, self-care, self-efficacy [36], resilience, self-acceptance, and pride [20] improve the well-being of LGBTIQ+ adolescents. In addition, positive coping skills increase well-being and self-esteem [63], healthy habits reduce suicide [73], as does help-seeking [5,57], which in turn increases well-being [40] and reduces depression [57]. The existing literature provides strong support for these conclusions, but it is crucial to acknowledge both the intricate interactions between environmental, psychological, and societal elements that influence mental health [77] and the individual factors that must be considered in clinical intervention. Spirituality also reduces depression and suicide and raises well-being [3,12,72,73]. Other factors that increase well-being are chest binding [43] and being out [60]. These results are relevant here, since some authors found that coming out can be a protective factor in finding a positive response, meaning that it depends on the context and the outcome. In McKay and Watson’s study (2020), coming out to all health professionals can be a protective factor against depression and boost self-esteem [60]. These findings contradict those of other studies, which suggest that coming out early in life can increase the likelihood of bullying and rejection from the family. As such, a thorough investigation of this element is needed [78]. These results partially support the third and last hypothesis: “individual protective factors are self-esteem and self-efficacy”, because initially, coming out or chest binding were not considered possible protective factors, but they have been proven to be so.

The less studied factors, part of the exosystem and macrosystem, emphasized the importance of access to health and counseling services specializing in affirmative therapy or medical care, which reduce anxiety [39] and depression [38] and increase well-being [20,41,61]. Additionally, having a good financial status decreases depression [70]. Regarding this factor, most of the studies selected were performed in and around the United States, where access to therapy and other affirming services must be paid for, meaning that only those whose family income is sufficiently high have access to these treatments. In addition, the political will to provide resources and policies aimed at the LGBTIQ+ community, including affirming care for adolescents, can improve health outcomes and reduce suicide attempts, especially for bisexual, questioning, and gay adolescents [61,68]. Despite the anti-trans climate and the dismantling of anti-discrimination policies for LGBTIQ+ people occurring in many countries, the specialist literature agrees that inclusive policies at school, combined with teacher training on inclusion policies, promotes coming out, life satisfaction, and a supportive school environment. In addition, these policies have an impact on reducing bullying and social stigma [79]. In short, an affirming political and professional context is key, and providing a situation where adolescents can come out and feel supported is an important protective factor [62].

## 5. Conclusions

From the standpoint of protective factors, this systematic evaluation offers a perceptive approach. Some factors in the social-ecological system as a whole are protective of the mental health and well-being of LGBTIQ+ adolescents, such as self-care, self-efficacy, resilience, self-acceptance, pride, family acceptance, social support from a romantic partner, gay-straight alliances, involvement in LGBTIQ+ activism, coping skills, emotion regulation strategies, help-seeking, affirmative therapy, access to counseling services, financial status, and policies and resources for LGBTIQ+ individuals, among others. Although it would be helpful to explore additional potential influences across domains, this study focuses on factors that affect mental health. 

The consideration of health variables is not a common strategy, as research has tended to emphasize risk factors. Moreover, most studies concentrate on adult LGBTIQ+ communities. Additionally, many of the articles examined come from the United States and, thus, must be interpreted within the framework of that nation and its culture. Additional research on the different adolescent substages, along with other life stages such as childhood or aging adults, could also provide new insights into the impact of protective factors. More reviews and studies are needed to improve community and therapeutic support according to APA guidelines [80] and, more importantly, to promote effective public policies in terms of mental health and social justice. In addition, professionals working with LGBTIQ+ youth should be trained specifically in the promotion of protective factors and how to change their intervention perspective according to these findings.

Lastly, according to the findings of this study, it is important to change the intervention model with LGBTIQ+ young people, promoting pro-LGBTIQ+ environments in schools and peer relationships. Another area of intervention involves family programs that can help parents understand their children, encouraging support and thus improving their mental health. To that end, rather than focusing on individual factors, a more structural approach that addresses all these aspects is needed.

## 6. Limitations

This study is not without limitations. First, it does not address all the existing protective factors but only those related to mental health (not, for instance, drug addiction or other addictions). Moreover, very few articles were found about European countries, and consequently, the results cannot be generalized to these populations. Additionally, most articles use convenience sampling strategies involving young people from a single city or urban area. These approaches limit the generalizability of the study’s findings, raising concerns about who is and is not represented in the sample of the general population. Because LGBTIQ+ youth are a hard-to-reach population, convenience sampling is often used in research involving them. Other articles that might be relevant to this paper had to be removed because they did not meet the risk of bias criteria [81,82]. Another limitation is that some items had problems identifying trans people, as it was not possible to determine whether they actively identified as LGBTIQ+ or were questioning their gender identity or orientation. This issue arises from the wording of the sociodemographic data items.

Future research should focus on protective factors related to the microsystem, exosystem, and macrosystem, as there is less research in these areas. Studies should seek to understand the context in which these protective factors develop. Multivariate analyses with direct, mediating, and moderating effects are required to explore these questions. More longitudinal studies are also needed to examine the long-term effects of protective factors for LGBTIQ+ adolescents. Additionally, as a diverse population group, LGBTIQ+ youth might have protective factors for their mental health and well-being that future research could uncover. Psychosocial factors can have different effects depending on age or developmental stage (early, emerging, middle, and late adulthood), gender (gay men versus lesbians), and sexual orientation (homosexual versus bisexual). An intersectional approach may prove beneficial for future studies, since protective factors may differ among youth who identify as LGBTIQ+ based on how their sexual orientation intersects with other oppressive social systems.

## Figures and Tables

**Figure 1 healthcare-12-01865-f001:**
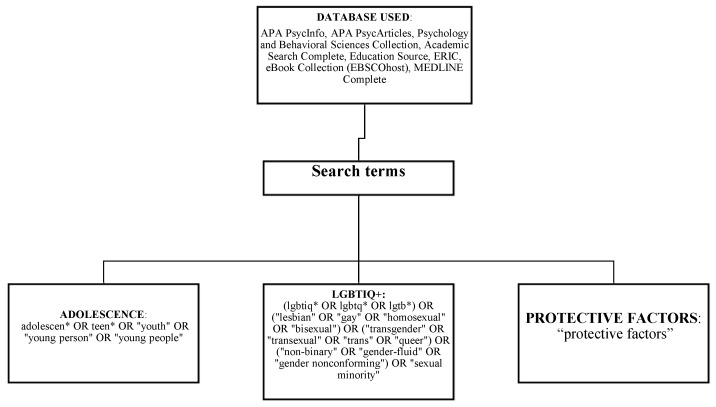
Search strategy: domain and search terms. The * refers to a symbol used in databases for searching words with different endings.

**Figure 2 healthcare-12-01865-f002:**
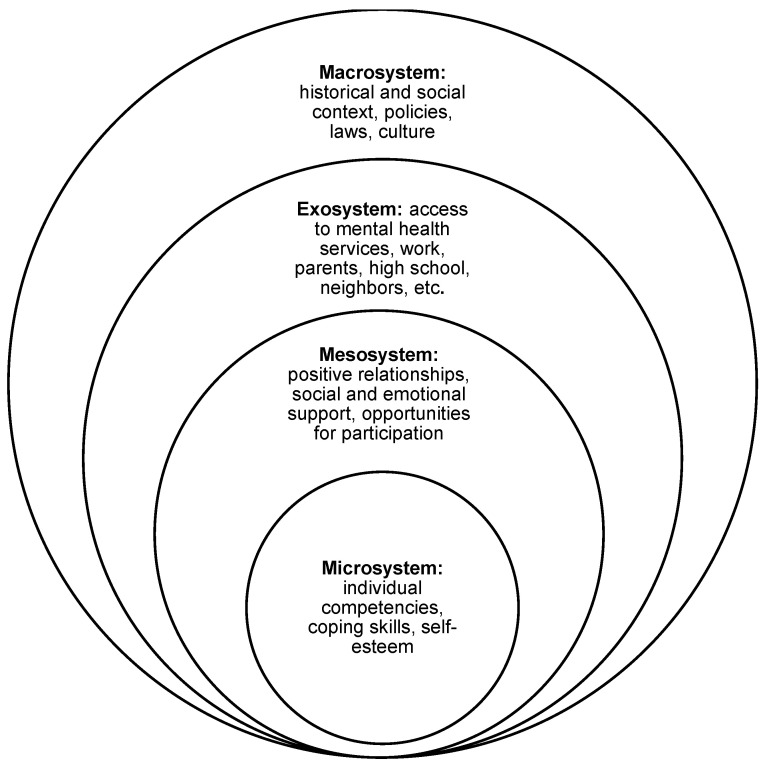
Socioecological model of protective factors.

**Figure 3 healthcare-12-01865-f003:**
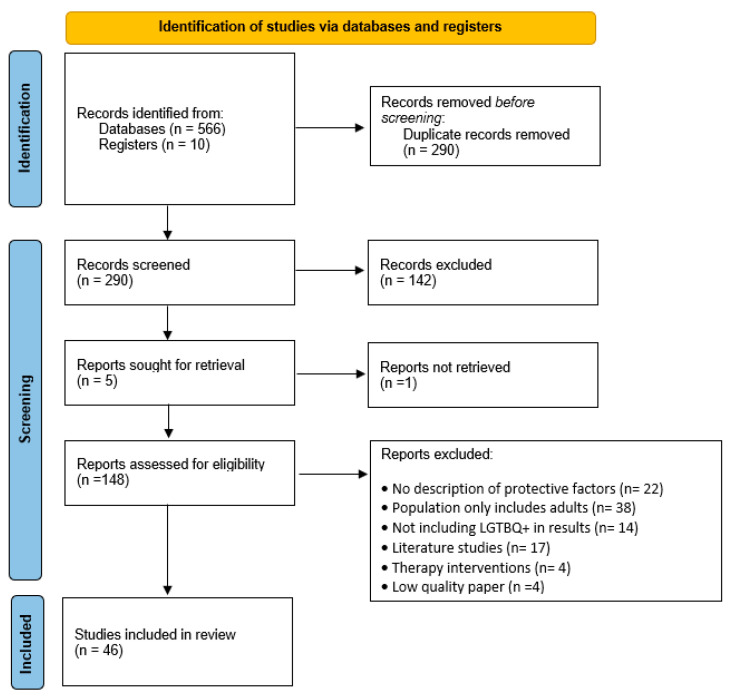
Preferred Reporting Items for Systematic Reviews and Meta-Analyses (PRISMA) study selection flow diagram outlining the literature review process when searching for articles. Adapted from Liberati et al. [25].

**Table 1 healthcare-12-01865-t001:** Mixed methods divided by socioecological model. The symbol √ indicates that it is present, while × signifies that it is not present.

Number, Authors and Year, Citation	Sampling Strategy		Study Design		N	Age	Gender Identity								Sexual Orientation							Race/Ethnicity								
	Random	Incidental	Longitudinal	Transversal			Cis Men	Cis Women	Global Cisgender	Trans Men	Trans Women	Global Trans/Gender Diverse/Minority	Non Binary/Gender Queer	Other	Straight/Hetero	Gay	Lesbian	Bi/Pansexual	Questioning	Asexual	Other	White	Black/African	Hispanic/Latino	Asian	Native Hawaiian/Pacific Islander	American Indian/Alaska Native/Native	Middle Eastern	Mixed/Multiracial	Other
Microsystem and mesosystem																														
1. Wilhelm et al. (2021) [7]	×	√	√	×	9844	11–15	×	×	√	×	×	√	×	×	×	×	×	×	×	×	×	√	√	√	√	√	√	×	√	×
Mesosystem																														
2. Pariseau et al. (2019) [33]	×	√	×	√	54	8–18	×	×	×	√	√	×	×	×	×	×	×	×	×	×	×	×	×	×	×	×	×	×	×	×
3. Luthar et al. (2021) [34]	×	√	×	√	14,603	11–18	√	√	×	×	×	×	√	×	×	×	×	×	×	×	×	√	√	√	√	×	×	×	×	√
4. Cardona et al. (2023) [35]	√	×	×	√	53	9–18	×	×	×	√	√	×	×	×	×	×	×	×	×	×	×	√	×	×	×	×	×	×	×	√
Exosystem																														
5. Andrzejewski et al. (2023) [36]	×	√	√	×	366	10–20	√	√	×	×	×	√	√	×	√	√	√	√	√	√	√	√	√	√	√	×	×	√	√	√

**Table 2 healthcare-12-01865-t002:** Qualitative methods divided by socioecological model. The symbol √ indicates that it is present, while × signifies that it is not present.

Number, Authors And Year, Citation	Sampling Strategy		Study Design		N	Age	Gender Identity								Sexual Orientation							Race/Ethnicity								
	Random	Incidental	Longitudinal	Transversal			Cis Men	Cis Women	Global Cisgender	Trans Men	Trans Women	Global Trans/Gender Diverse/Minority	Non-Binary/Gender Queer	Other	Straight/Hetero	Gay	Lesbian	Bi/Pansexual	Questioning	Asexual	Other	White	Black/African	Hispanic/Latino	Asian	Native Hawaiian/Pacific Islander	American Indian/Alaska Native/Native	Middle Eastern	Mixed/Multiracial	Other
Mesosystem																														
6. Morgan et al. (2022) [37]	×	√	×	√	14	14–23	×	×	×	√	√	×	√	×	×	×	×	×	×	×	×	×	×	×	×	×	×	×	×	×
7. Johnson et al. (2020) [38]	×	√	×	√	24	16–20	√	√	×	√	√	×	√	√	×	×	×	×	×	×	×	√	√	√	√	×	×	√	√	×
Exosystem																														
8. Pampati et al. (2021) [39]	×	√	×	√	42	15–24	√	√	×	√	√	×	√	√	×	×	×	×	×	×	×	√	√	√	×	×	×	×	√	×
Microsystem and mesosystem																														
9. Bridge et al. (2022) [40]	×	√	×	√	20	16–24	√	√	×	×	×	×	√	×	×	√	√	√	√	√	√	√	√	×	√	×	×	×	√	√
Microsystem, mesosystem and exosystem																														
10. Miller et al. (2022) [20]	×	√	×	√	15	17–25	√	√	×	×	×	√	√	×	×	×	×	√	×	×	×	×	×	×	×	×	×	×	×	×
11. Greenfield et al. (2021) [41]	×	√	√	×	57	17–26	×	×	×	√	√	×	√	√	√	√	√	√	√	√	×	√	√	√	×	×	×	×	√	×

**Table 3 healthcare-12-01865-t003:** Quantitative methods divided by socioecological model. The symbol √ indicates that it is present, while × signifies that it is not present.

Number, Authors and Year, Citation	Sampling Strategy		Study Design		N	Age	Gender Identity								Sexual Orientation							Race/Ethnicity								
	Random	Incidental	Longitudinal	Transversal			Cis Men	Cis Women	Global Cisgender	Trans Men	Trans Women	Global Trans/Gender-Diverse/Minority	Non-Binary/Gender Queer	Other	Straight/Hetero	Gay	Lesbian	Bi/Pansexual	Questioning/Unsure	Asexual	Other/Sexual Minority	White	Black/African	Hispanic/Latino	Asian	Native Hawaiian/Pacific Islander	American Indian/Alaska Native/Native	Middle Eastern	Mixed/Multiracial	Other/Minority
Microsystem																														
12. Hatchel et al. (2019) [42]	√	×	×	√	713 (4867 *)	12–18	√	√	×	×	×	√	×	√	√	√	√	√	√	×	√	√	×	√	×	×	×	×	√	×
13. Julian et al. (2021) [43]	×	√	×	√	684	13–24	×	×	×	×	×	√	×	×	×	×	×	×	×	×	×	×	×	×	×	×	×	×	×	×
14. Chinazzo et al. (2023) [44]	×	√	×	√	166	13–25	×	×	×	√	√	×	√	×	×	×	×	×	×	×	×	×	×	×	×	×	×	×	×	×
15. Jadva et al. (2023) [45]	×	√	×	√	3713	11–19	√	√	×	×	×	√	√	×	√	√	√	√	√	×	√	√	√	×	√	×	×	×	×	√
16. Eisenberg et al. (2019) [46]	√	×	×	√	2168	14–17	×	×	×	√	√	×	×	×	×	×	×	×	×	×	×	√	√	√	√	×	√	×	√	√
Mesosystem																														
17. Hong et al. (2023) [47]	×	√	×	√	105 (338 *)	12–22	×	×	×	×	×	×	×	×	√	√	√	√	×	×	√	×	√	×	×	×	×	×	×	×
18. Vanbronkhorst et al. (2021) [12]	×	√	×	√	99 (334 *)	12–17	√	√	×	√	√	×	×	√	√	√	√	√	√	×	√	√	√	√	×	×	×	×	√	√
19. Whitton et al. (2021) [48]	×	√	√		488	16–20	√	√	×	×	×	√	√	×	×	×	√	√	×	×	√	√	√	√	×	×	×	×	×	√
20. Lardier et al. (2020) [49]	×	√	×	√	110 (687 *)	16–18	√	√	×	×	×	×	×	×	×	√	√	√	×	×	√	×	√	√	×	×	×	×	×	×
21. Garthe et al. (2020) [50]	×	√	√	×	78	16–29	√	√	×	×	√	×	×	√	√	√	√	×	×	×	√	√	√	√	√	×	√	×	√	√
22. Hong et al. (2021) [51]	×	√	×	√	105 (580 *)	13–24	√	√	×	×	×	×	×	×	√	√	√	√	×	×	√	×	√	×	×	×	×	×	×	×
23. Stroem et al. (2021) [52]	×	√	√	×	3296	14–15	×	×	√	×	×	√	×	×	√	×	×	×	×	×	√	√	√	√	×	×	×	×	√	√
24. Ross-Reed et al. (2019) [53]	√	×	√	×	858 (18,451 *)	14–18	√	√	×	√	√	×	×	×	×	×	×	×	×	×	×	√	√	√	√	√	√	×	×	×
25. Berona et al. (2021) [54]	×	√	√	×	1641	17–24	√	√	×	×	×	√	×	×	×	×	×	×	×	×	√	√	√	√	×	×	×	×	×	√
26. Horwitz et al. (2021) [55]	×	√	×	√	1275 (6423 *)	12–17	√	√	×	×	×	×	×	×	√	√	√	√	√	×	√	√	√	√	×	×	×	×	√	√
27. Wang et al. (2023) [56]	×	√	×	√	637 (14,265 *)	13–20	√	√	×	×	×	×	×	×	√	√	√	√	×	×	×	×	×	×	×	×	×	×	×	×
28. Semprevivo (2023) [57]	√	×	×	√	303 (1104 *)	14–18	√	√	×	√	√	×	×	×	×	√	√	√	√	×	×	√	√	√	√	√	√	×	√	×
29. Taliaferro et al., (2019a) [58]	√	×	×	√	2168	14–17	×	×	√	×	×	√	×	×	×	×	×	×	×	×	×	√	×	×	×	×	×	×	×	√
30. Pollitt et al. (2021) [59]	×	√	√	×	129	15–21	×	×	×	√	√	×	×	√	√	√	√	√	√	×	√	√	√	×	√	√	×	×	√	√
31. McKay and Watson (2020) [60]	×	√	×	√	3624	13–17	√	√	×	√	√	×	√	×	√	√	√	√	√	√	√	√	√	√	√	×	×	√	√	√
Exosystem																														
32. Chan and Leung (2023) [61]	×	√	×	√	793	12–18	√	√	×	×	×	×	√	×	√	√	√	√	√	√	√	×	×	×	√	×	×	×	×	√
33. Ioverno (2023) [62]	×	√	×	√	66,851	15–24	√	√	×	√	√	×	√	×	√	√	√	√	×	×	√	√	×	×	×	×	×	×	×	√
Microsystem and mesosystem																														
34. Parmar et al. (2022) [63]	√	×	√	×	1943 (14,317 *)	11–34	×	×	×	√	√	×	×	×	×	×	×	×	×	×	×	√	√	√	√	√	√	×	×	√
35. Conn et al. (2023) [64]	×	√	×	√	315	12–20	×	×	×	√	√	×	√	√	×	×	×	×	×	×	×	√	√	√	√	√	×	×	√	√
36. Greenspan et al. (2023) [65]	×	√	×	√	31	13–17	×	×	√	×	×	√	√	√	√	√	√	√	√	√	√	√	√	×	×	×	×	×	√	×
37. McPherson et al. (2023) [66]	√	×	×	√	3861	12–17	×	×	×	×	×	√	×	√	×	×	×	×	×	×	×	√	√	√	√	×	√	×	√	×
38. Zell and Kerr (2024) [67]	√	×	×	×	6591 (26,887 *)	13–17	√	√	×	×	×	√	×	×	√	×	×	×	×	×	√	√	√	√	×	√	√	×	√	√
Mesosystem and exosystem																														
39. Eisenberg et al. (2021) [68]	√	×	×	√	2409 (20,790 *)	M = 16	√	√	×	×	×	×	×	×	×	√	√	√	√	×	×	√	√	√	√	×	×	×	√	√
40. Valido et al. (2021) [69]	√	×	×	√	4822	12–18	√	√	×	√	√	×	×	×	√	√	√	√	×	×	×	√	√	√	√	√	√	×	√	√
41. Rider et al. (2022) [5]	√	×	×	√	12,750 (67,806 *)	14–16	√	√	×	√	√	×	√	√	√	√	√	√	√	×	√	√	√	√	√	×	√	×	√	×
42. Burstein et al. (2023) [70]	×	√	×	√	562/540 (16,288/10,792 *)	14–18	×	×	√	×	×	√	×	×	×	×	×	×	×	×	×	√	×	×	×	×	×	×	×	√
Microsystem, mesosystem and exosystem																														
43. Taliaferro et al. (2019b) [71]	√	×	×	√	1635	14–17	×	×	×	×	×	√	×	×	×	×	×	×	×	×	×	√	√	√	√	√	√	×	√	×
44. Mintz et al. (2021) [72]	√	×	√	×	580 (2744 *)	12–18	√	√	×	×	×	√	×	×	√	√	√	√	√	×	√	√	√	√	√	√	√	×	√	×
45. Rivas-Koehl et al. (2021) [73]	√	×	×	√	1078	12–18	√	√	×	×	×	√	×	√	×	√	√	√	√	×	√	√	√	√	√	√	√	×	√	×
46. Valido et al. (2022) [3]	×	√	×	√	4778	14–16	√	√	×	×	×	√	×	√	√	√	√	√	√	×	√	√	√	√	√	√	√	×	√	

* N_total_.

**Table 4 healthcare-12-01865-t004:** Protective factors and outcomes based on study assigned number. The symbol √ indicates that it is present, while × signifies that it is not present.

	Compare cis-LGBT	Less Depression	Less Anxiety	Less Suicide Ideation and Attempts	Less NSSI	Less Internalizing and Externalizing Problems	Fewer Stressors or Less Stress Management	Fewer Psychosomatic Symptoms/Fewer MH Outcomes	Less Bullying/Sexual Violence Victimization/Harassment and Violence Perpetration	Sexual Violence and Dating Violence/Harassment	Well-Being/MH Access	Self-Esteem/Self-Acceptance	Life Satisfaction	Feeling Safe/ Connectedness/ Belonging	Other
Microsystem															
Manage stress	×	×	×	×	×	×	×	×	×	×	×	9	×	9	×
Care about school achievement/learning experiences	3	×	×	43	43	×	×	×	×	×	3	×	×	×	×
Stay out of conflict and manage conflict	1	×	×	×	×	×	×	×	×	×	×	×	×	×	×
Help-seeking beliefs/help-seeking most time and from adult(s)	28	28, 44	×	12, 28, 44, 45	×	×	×	×	44	44	×	×	×	×	×
Chest binding	×	×	×	×	×	×	×	×	×	×	13	×	×	×	×
Personal resilience											10				
Self-care, self-acceptance (including sexuality), self-esteem, gender positivism, and pride	×	35	×	×	×	×	×	34	×	×	10, 14, 35, 36	9, 36	×	×	×
Relying on oneself	×	×	×	×	×	×	×	×	11	×	×	×	×	×	11-self-protection
Internal developmental assets/positive coping skills/problem solving/self-efficacy	×	16, 37	37	16, 37, 38	16, 37	×	×	34	×	×	×	×	×	×	×
Emotional well-being	×	×	×	×	×	×	×	34	×	×	×	×	×	×	×
Healthy activities	×	×	×	45	×	×	×	×	×	×	×	×	×	×	×
Spirituality	40, 46	44	×	44, 45	×	×	×	×	40, 44, 46	44, 46	×	×	×	×	×
Mesosystem															
Parental support/trust/communication	1, 23, 29	7, 21, 35	21	7, 26, 27, 43	43	×	×	7	23	×	6, 7, 29, 35	6, 7, 21	6, 7	×	6-stable housing 7-lower perceived burden
Family support/cohesion	22, 24, 40, 46	2, 37, 44	2, 37	2, 18, 20, 24, 26, 37, 44, 45	24, 37	2	×	22, 34	20, 40, 44, 46	24, 40, 44, 46	10, 36	×	36	×	×
School support/importance/teacher support/feel safe at school	3, 17, 22, 24, 29, 41	37, 41, 42	37	15, 20, 24, 37, 41, 45	15, 24, 37,41	×	×	17, 22, 34	20	24	3, 10, 29, 36	×	36	17	×
Peer/friend support	24, 29, 40, 46	4, 21	4, 21	24, 45	24	×	×	×	40, 46	24, 40, 46	10, 29	21	×	×	×
Social support/connections	28	28	×	20, 25, 28	25	×	×	×	20, 25	×	×	9	×	×	×
Adult support/caring	29, 46	42, 44	×	38, 43, 44	43	×	×	×	44, 46	44, 46	29	×	×	×	×
Community support/feel that they matter to community	24, 28	28, 42	×	18, 24, 28,	24	×	×	×	×	24	10, 36	×	36	×	×
Respect/helping others	1, 40	×	×	×	1	×	×	×	40	×	×	×	×	×	×
GSA	5, 39	5		39							5				
Positive models/spiritual leaders	×	×	×	18	×	×	×	×	×	×	10	×	×	×	×
Relying on others	×	×	×	×	×	×	×	×	11	×	×	×	×	×	11-self-protection
Romantic involvement	×	×	×	×	×	×	×	×	×	×	19	×	×	×	×
Trans collective self-esteem	×	21	21	×	×	×	×	×	×	×	×	21	×	×	×
Environment where the chosen name is used (family, school, work)	×	30	×	30	×	×	×	×	×	×	×	30	×	×	×
Being out to health care providers	×	31	×	×	×	×	×	×	×	×	×	31	×	×	×
Exosystem and macrosystem															
Access and communication from health care/mental care services	40, 41, 46	41	8	41, 45	41	×	11	×	40, 46	46	11	×	×	×	×
Access to affirming counseling experiences	40, 46	44	×	44, 45	×	×	11	×	40, 44, 46	40, 44, 46	10, 11	×	×	×	×
Collective action	×	×	×	×	×	×	×	×	32	×	32	×	×	×	×
LGBTIQ+ community resources	39	×	×	39	×	×	×	×	×	×	×	×	×	×	×
Inclusive and safety school policies, concealment, inclusive teacher/staff training and curricula	39	33	×	39, 43	43	×	×	×	33	33	×	×	×	33	33-being out at school
Good financial status	×	42	×	×	×	×	×	×	×	×	×	×	×	×	×

## Data Availability

The raw data supporting the conclusions in this article will be made available by the authors, without undue reservation, to any qualified researcher.

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
