# Peer review of "Protective Factors in the LGBTIQ+ Adolescent Experience: A Systematic Review"

_healthcare, 2024, doi:10.3390/healthcare12181865_

Round 1
Reviewer 1 Report
Comments and Suggestions for Authors
This is a very interested topic, while the paper is well organised and written. The methodology is thorough and complete, which enhances the study’s scientific rigor. The authors should revisit the following important issues:
1. Justification needs to be improved. Why is this study necessary? What gaps does it have to fill in? How does it differ from other systematic reviews on the subject? For example, this one https://pubmed.ncbi.nlm.nih.gov/28394718/
2. Discussion is poor. It reads like notes. It needs a more thorough and detailed discussion of the findings in relation to the socioecological model. Also, any recommendations for improving the protection of adolescents who identify themselves as LGBTIQ+? Maybe a framework or a new model of action? What about any limitations of the study?
3. How do you know that you have identified all relevant studies? This is because in your search keywords you have not included the health or psychosocial outcomes. For example, you have not included keywords such as ‘suicide’, or ‘social isolation’? Maybe there are more studies which have used other terms than ‘protective factors’. It seems that you have not exhausted all possibilities with your choice of keywords.
Author Response
Thank you very much for taking the time to review this manuscript. Please find the detailed responses below and the corresponding revisions/corrections highlighted/in track changes in the re-submitted files. Please, see the attachement for more extensive information.
Comments 1: Justification needs to be improved. Why is this study necessary? What gaps does it have to fill in? How does it differ from other systematic reviews on the subject? For example, this one https://pubmed.ncbi.nlm.nih.gov/28394718
Response 1: We agree that the introduction needs more work. We have improved the article’s justification in the introduction.
Comments 2: Discussion is poor. It reads like notes. It needs a more thorough and detailed discussion of the findings in relation to the socioecological model. Also, any recommendations for improving the protection of adolescents who identify themselves as LGBTIQ+? Maybe a framework or a new model of action? What about any limitations of the study?
Response 2: Thank you for your comment. We fully agree that the discussion required more development. We have broadened the discussion and conclusions, including some recommendations for improving the protection of adolescents, and we added some information regarding the limitations of the study.
Comments 3: How do you know that you have identified all relevant studies? This is because in your search keywords you have not included the health or psychosocial outcomes. For example, you have not included keywords such as ‘suicide’, or ‘social isolation’? Maybe there are more studies which have used other terms than ‘protective factors’. It seems that you have not exhausted all possibilities with your choice of keywords.
Response 3: We have justified the perspective of using “protective factors” as a key word in the methodology.
Reviewer 2 Report
Comments and Suggestions for Authors
This article is a well-conducted systematic review that provides important insights into the protective factors that support the mental health and well-being of LGBTIQ+ adolescents.
However, it is necessary to clarify the limitations of the studies reviewed. This aspect is crucial for a rigorous interpretation of the relationships identified by the authors. For example, in Table 4, the authors present the protective factor "Care about school achievement/learning experiences" as having an impact on "Less suicide ideation and attempts" and "Less NSSI," based on the study by Taliaferro et al. (2019) "Risk and Protective Factors for Self-Harm in a Population-Based Sample of Transgender Youth." However, the cited study clarifies that limitations may compromise the interpretation of the data:
"One limitation involved some of the survey measures. The measure assessing transgender/GNC identity represented a weakness, as the item wording did not permit us to distinguish between students who were unsure of their gender identity and those who actively identify as transgender/GNC. The item assessing suicide attempts did not include a definition of the behavior; thus, some students may have misinterpreted some of their suicide-related behaviors as attempts."
Therefore, I consider it necessary to clarify for readers the limitations of the research on which the authors base their conclusions.
Author Response
Thank you very much for taking the time to review this manuscript. Please find the detailed responses below and the corresponding revisions/corrections highlighted/in track changes in the re-submitted files
Comments 1: However, it is necessary to clarify the limitations of the studies reviewed. This aspect is crucial for a rigorous interpretation of the relationships identified by the authors. For example, in Table 4, the authors present the protective factor "Care about school achievement/learning experiences" as having an impact on "Less suicide ideation and attempts" and "Less NSSI," based on the study by Taliaferro et al. (2019) "Risk and Protective Factors for Self-Harm in a Population-Based Sample of Transgender Youth." However, the cited study clarifies that limitations may compromise the interpretation of the data:
"One limitation involved some of the survey measures. The measure assessing transgender/GNC identity represented a weakness, as the item wording did not permit us to distinguish between students who were unsure of their gender identity and those who actively identify as transgender/GNC. The item assessing suicide attempts did not include a definition of the behavior; thus, some students may have misinterpreted some of their suicide-related behaviors as attempts."
Therefore, I consider it necessary to clarify for readers the limitations of the research on which the authors base their conclusions.
Response 1: We have elaborated on the limitations of the study in the conclusion section (Pages 18-19)
Reviewer 3 Report
Comments and Suggestions for Authors
This is an interesting and rich subject. Here are some thoughts which occurred to me, on reading it, which might be useful:
1. While this is an intriguing idea, it is not well executed. The topic treats "Protective factors in the LGBTIQ+ adolescent experience". However, for the article to reach publication standard there would have to be a far more critical approach to the underpinning concepts of the LGBTIQ+ adolescent experience. Authors should thoroughly examine the potential differences in ego development and self-experience during adolescence between heterosexual and LGBTIQ+ individuals. This includes considering factors such as coming out, social stigma and bullying, as well as the challenges of adapting or facing rejection from family and friends.
2. Authors failed to approach their findings from a developmental standpoint.
3. Authors did not include any limitations of their study
4. This topic has several implications for future research and clinical practice, however, these have not been described or discussed.
5. The discussion section in general leaves the reader wanting more detail.
Author Response
Thank you very much for taking the time to review this manuscript. Please find the detailed responses below and the corresponding revisions/corrections highlighted/in track changes in the re-submitted files. Please see the file for more extensive information.
Comments 1: While this is an intriguing idea, it is not well executed. The topic treats "Protective factors in the LGBTIQ+ adolescent experience". However, for the article to reach publication standard there would have to be a far more critical approach to the underpinning concepts of the LGBTIQ+ adolescent experience. Authors should thoroughly examine the potential differences in ego development and self-experience during adolescence between heterosexual and LGBTIQ+ individuals. This includes considering factors such as coming out, social stigma and bullying, as well as the challenges of adapting or facing rejection from family and friends.
Response 1: We have developed the discussion in depth by focusing on various aspects that the reviewer mentioned.
Comments 2: Authors failed to approach their findings from a developmental standpoint
Response 2: We have briefly elaborated on adolescent self-development, discussing identity construction and minority stress.
Comments 3: Authors did not include any limitations of their study
Response 3: We have elaborated on the limitations of the study in the discussion section. Thank you for your suggestion.
Comments 4: This topic has several implications for future research and clinical practice, however, these have not been described or discussed.
Response 4: The implications for future research and clinical practice have been included in the discussion section.
Comments 5: The discussion section in general leaves the reader wanting more detail.
Response 5: We have expanded the discussion section. Thank you for your comment.

Round 2
Reviewer 1 Report
Comments and Suggestions for Authors
My comments have been addressed. The paper is in better shape now. Thank you for considering my feedback.
Author Response
Thank you.
Reviewer 2 Report
Comments and Suggestions for Authors
The authors corrected the lack of reference to the study's limitations.
Author Response
Thank you